# The Fatty Acid and Mineral Composition of Cobb 500 Broiler Meat Influenced by the Nettle (*Urtica dioica*) Dietary Supplementation, Broiler Gender and Muscle Portion

Nikola Stanišić *[ID], Zdenka Škrbić, Veselin Petričević, Danijel Milenković [ID], Maja Petričević [ID], Marija Gogić [ID] and Miloš Lukić [ID]

Institute for Animal Husbandry, Belgrade—Zemun, Autoput 16, P.O. Box 23, 11080 Belgrade, Serbia
* Correspondence: nikola0135@yahoo.com

**Abstract:** The objective of the present trial was to evaluate the effects of nettle leaves (*Urtica dioica*) supplementation, broiler gender and muscle portion on meat fatty acid and mineral profiles. Prior to the trial, a total of 600 one-day-old Cobb 500 broiler chickens of both genders, equally, were randomly divided into three groups: a basal diet (control group) and a basal diet supplemented in the last two weeks of fattening with fresh nettle leaves (30 g/kg diet) or with oven dry nettle leaves (5 g/kg diet). After reaching 42 days of age, twenty birds per diet group were randomly selected (four birds per box, both genders equally) and slaughtered, and breasts and drumsticks were collected for analysis. The dry nettle supplementation increased the MUFA and lowered the PUFA, n-6 and n-6/n-3 ratio, mainly in female broilers ($p < 0.05$). Drumsticks had higher MUFA and PUFA and lower SFA and n-6/n-3 ratio compared to breast meat ($p < 0.05$). The addition of nettle to broiler diet increased Fe, Zn and Se and decreased Ca levels, mainly in drumsticks ($p < 0.05$). Furthermore, drumstick meat had more elements with proven health benefits, such as Fe, Zn, Ca and Mn, than breast meat. This study suggests that feeding Cobb 500 broilers with the addition of nettle leaves can significantly change meat fatty acid and mineral composition, but this effect differed between gender and portion.

**Keywords:** broiler meat; nettle; gender; muscle portion; fatty acids; minerals

## 1. Introduction

There is a trend in present animal production to replace specific feed additives, such as chemical growth promoters and antibiotics, mainly resulting from the ban on using these supplements [1]. The ingredients most used for this purpose are natural biologically active compounds such as probiotics, prebiotics, organic acids and herb mixtures [2,3]. Specific herbs, known as natural phytogenic feed additives with antimicrobial and antioxidant effects, such as garlic, cinnamon, anise, coriander, oregano, pepper, chili, nettle, rosemary, thyme and many others, can potentially be used as farm animal performance enhancers [4–6]. The majority of studies on the effects of various herbs on broiler performance have reported positive effects on gain, feed conversion and carcass quality [7–13].

Stinging nettle (*Urtica dioica*, of the family of Urticaceae) can be considered a phytogenic feed additive in the poultry diet [14], as it is reported to have numerous health benefits and many biochemical effects, such as antioxidant, antibacterial and antifungal [15,16]. Previous studies have shown that nettle leaves are rich in vitamins A and C, minerals Fe, Mg, Mn, Zn and Se, and polyunsaturated fatty acids, making them particularly suitable as a chicken feed additive [17–19]. It is further hypothesized that it may cause a change in broiler blood lipid and intramuscular fatty acid profile [20], as it has been reported to influence lipid metabolism and lipoprotein synthesis [15,19].

Studies of nettle as a feed additive in broiler production mainly show positive effects on performance and carcass traits. In several experiments, a level of 0.75 to 2% nettle added to the diet had a positive effect on body weight gain and carcass traits of broiler

chickens [9,19,21–23]. The best results when implementing different levels of nettle, in the research of Bekele et al. [24], are obtained with the rate of 9% of nettle leaves in the diet, with reported improved feed intake and daily weight gain. On the other hand, in the research of Nasiri et al. [17], different nettle levels in the starter and growing feeds show no significant effect on feed intake, weight gain and feed conversion of broilers. Similarly, Keshavarz et al. [25], have found that broiler performance is not altered by the dietary inclusion of 1% nettle powder or essential oil.

Even though the previous investigations on *Urtica dioica* as a feed supplement show the impact on broiler performance, the data on the meat quality could be more extensive. Most of these studies have reported improved antioxidant capacity of meat [15,19,26]. However, in the research of Keshavarz et al. [25], nettle shows no influence on the oxidative stability of meat. Safamehr et al. [23] report that serum triglyceride and cholesterol concentrations were significantly decreased in broilers a fed diet containing 1% of nettle. In contrast, Khosravi et al. [27] find no significant effect of feeding nettle extract on total cholesterol. According to the study of Đukić Stojčić et al. [28], fresh nettle leaves supplementation (40 and 80 g per bird/daily) changes the fatty acid profile by increasing MUFA and n-6 PUFA and lowering n-3 PUFA levels of chicken breast meat.

To the best of our knowledge, there are insufficient data on the effect of nettle supplementation of broiler diet on meat fatty acid composition, as well as no data on the mineral composition of broiler meat. Therefore, the objective of the present study was to test if dietary supplementation with nettle leaves in the last two weeks of six-week fattening period, significantly affects these two quality attributes of Cobb 500 broiler meat. This experiment considered the effects of dry and fresh nettle leaves, and the effects of factors such as broiler gender and the muscle portion of broilers. Data obtained in this study will not only provide an understanding of the effect of nettle on meat quality but will also provide insights into factors such as gender and the muscle portion, which may influence meat composition.

## 2. Materials and Methods

### 2.1. Stinging Nettle (Urtica dioica) Leaf Meal Preparation

Stinging nettle was harvested in May 2021 (spring season) at the Institute for Animal Husbandry (Belgrade, Serbia). After collecting, sorting and washing, only the leaf part was used in the experiment. The collected leaves were partially used as such (fresh), and partially as oven dried (70 °C for 15 h), and ground to a powder using a coffee grinder. Both nettle samples (fresh and dried) were sealed in polyethylene bags and stored in cooled conditions before being mixed into the standard diet. Standard chemical analyses showed that fresh leaves contained 85.0% moisture, 3.0% ash, 0.6% fat and 4.6% protein, and oven dried leaves contained 3.4% moisture, 19.2% ash, 3.4% fat and 27.8% protein. The fatty acid composition (% of total fatty acids) of nettle used in this trial was 4.91 and 7.90% (C16:0), 2.43 and 3.05% (C16:1 n-7), 4.51 and 4.70% (C18:0), 5.43 and 4.71% (C18:1 n-9), 21.63 and 16.60% (C18:2 n-6), and 8.32 and 5.59% (C18:3 n-3) for fresh and dry leaves, respectively. The mineral composition (mg/100 g dry basis) of nettle used in this trial was 201.36 and 306.5 mg (Ca), 29.9 and 32.6 mg (Fe), 3.1 and 3.4 mg (Na), 6.92 and 7.26 mg (Mg), 3.5 and 3.8 mg (Zn), 1255 and 1277 mg (K), and 2.5 and 2.6 mg (Mn) for fresh and dry leaves, respectively.

### 2.2. Animal Management and Experimental Design

The study protocol was approved and conducted following the Animal Ethics Committee guidelines of the Institute for Animal Husbandry (Belgrade, Serbia). The experiment was performed on 600 one-day-old Cobb 500 hybrid broiler chickens of both genders (43.2 g average initial body weight). The chickens were housed in a floor system with chopped straw litter (dry chopped wheat straw 2–4 cm long), in 15 boxes (5 boxes per dietary treatment) and 40 birds per box (approx. 0.1 m$^2$ space allowance per bird). The gender ratio was equal and the same in each box. Water and feed were provided ad libitum, and the

light and temperature programs used in the experiment were set according to the ambient recommendations of the manufacturer of the used hybrid. All groups were fed the standard diet presented in Table 1, whereas the experimental groups received additional nettle leaves from the 29th day of feeding. The amount of nettle added to experimental groups was calculated according to Shonte et al. [29], who state that the nutrient composition increases approximately six times by oven-drying fresh nettle leaves. The trial was a completely randomized design, where birds were allocated into three groups as follows:

- Control group = standard diet.
- Fresh nettle group = standard diet + 3.0% coarse grounded fresh nettle leaves.
- Dry nettle group = standard diet + 0.5% fine grounded dry nettle leaves.

**Table 1.** Feed composition (g/kg) and calculated nutrient content (g/100 g) of the basal diets used in the trial.

| Ingredient (g/kg) | Starter (0–21 d) | Finisher (22–42 d) |
|---|---|---|
| Corn | 532 | 593 |
| Soybean meal (44% CP) | 310 | 240 |
| Soybeans (full fat, extruded) | 80 | 100 |
| Soybean oil | 20 | 20 |
| Monocalcium phosphate | 20 | 15 |
| Limestone | 14 | 11 |
| Vitamin + mineral supplement [1] | 10 | 10 |
| L-lysine | 4 | 3 |
| DL-methionine | 3 | 2 |
| L-threonine | 2 | 1 |
| Salt | 3 | 3 |
| Mycotoxin binder (zeolite) [2] | 2 | 2 |
| Total | 1000 | 1000 |
| Nutrients and energy level (calculated) | | |
| ME (MJ/kg) | 12.5 | 13.0 |
| Crude protein (%) | 21.3 | 19.2 |
| Crude fat (%) | 5.85 | 6.33 |
| Crude fibre (%) | 3.88 | 3.70 |
| Lysine, digestible (%) | 1.35 | 1.15 |
| Methionine, digestible (%) | 0.62 | 0.50 |
| Calcium (%) | 0.96 | 0.83 |
| Phosphorus, available (%) | 0.49 | 0.40 |

[1] Contained per kg of diet: Vitamin A 1000 IJ; thiamine 3 mg; riboflavin 9 mg; nicotinic acid 60 mg; pantothenic acid 15 mg; pyridoxine 4 mg; folic acid 2 mg; vitamin $B_{12}$ 0.02 mg; biotin 0.15 mg; choline 500 mg; vitamin $D_3$ 5000 IJ; vitamin E 50 mg; vitamin $K_3$ 3 mg; Mn 100 mg; Fe 40 mg; Zn 110 mg; Cu 15 mg; I 0.5 mg; Se 0.3 mg. [2] Technological additive based on clinoptilolite with the commercial name MINAZEL® (Patent Co, Mišićevo, Serbia).

On day 42, twenty birds per diet group were randomly selected (four birds per box, both genders equally) and slaughtered under commercial conditions in the Experimental slaughterhouse of the Institute for Animal Husbandry (Belgrade, Serbia). After chilling (at 4 °C for 24 h), meat samples from breasts and drumsticks were collected from each carcass, sealed in plastic bags under vacuum and stored at −20 °C until analyses were performed. Fatty acid and mineral analyses were performed on each meat sample after they had been thawed at +4 °C a day before, and homogenized in a commercial blender (Bosch MMB6141B).

*2.3. Fatty Acid Analysis*

Fatty acid methyl esters (FAMEs) were prepared by direct transesterification, as described by O'Fallon et al. [30]. In brief, the sample (1 g of meat) was placed into a 15 mL tube to which 1.0 mL of the methanol ($CH_3OH$), 0.7 mL of 10 N KOH in water, and 5.3 mL of MeOH were added. The tube was incubated in a 55 °C water bath for 1.5 h with vigorous

manual shaking for 5 s every 20 min. After cooling below room temperature in a cold tap water bath, 0.58 mL of 24 N $H_2SO_4$ in water was added. The tube was mixed by inversion and, with precipitated $K_2SO_4$ present, incubated in a 55 °C water bath for 1.5 h and shaken by hand for 5 s every 20 min. After FAME synthesis, the tube was cooled in a cold tap water bath, and 3 mL of hexane was added. After shaking, the organic phase, including the FAMEs, was separated and subjected to GC analysis.

The GC instrument Shimadzu 2014 (Kyoto, Japan), used for FAMEs determination, was equipped with a split/splitless injector, fused silica cianopropyl HP-88 column (length 60 m, i.d. 0.25 mm, film thickness 0.20 μm) and flame ionisation detector (FID). The column temperature was programmed. The injector and detector temperatures were 260 and 280 °C, respectively. The carrier gas was helium at a flow rate of 1.0 mL/min and injector split ratio of 1:10. Injected volume was 1 μL. The GC oven program started at 50 °C (hold time 2 min), which then was raised at 20 °C/min to 190 °C, at 10 °C/min to 200 °C (hold time 10 min), and at 15 °C/min to 250 °C (hold time 2 min). Chromatographic peaks in the samples were identified by comparing relative retention times of FAMEs peaks with peaks in a Supelco 37 Component FAMEs mix standard (Sigma-Aldrich, Hamburg, Germany) and individual fatty acids. The concentration of each fatty acid was expressed as a percentage of total FA.

### 2.4. Mineral Analysis

The analysis of the following isotopes: sodium ($^{23}$Na), magnesium ($^{24}$Mg), potassium ($^{39}$K), calcium ($^{44}$Ca), manganese ($^{55}$Mn), iron ($^{57}$Fe), zinc ($^{66}$Zn) and selenium ($^{77}$Se) was performed by "iCap Q" ICP-MS (Thermo Scientific, Bremen, Germany), equipped with a collision cell and operating in kinetic energy discrimination (KED) mode in order to minimise polyatomic-based Ar2 interferences. Before analyses, aliquots of approximately 0.3 g of homogenised meat samples were transferred into Teflon vessels, 5 mL nitric acid (67% Trace Metal Grade, Fisher Scientific, Loughborough, UK) and 1.5 mL hydrogen peroxide (30% analytical grade, Sigma-Aldrich, USA) were added. The microwave (Start D, Milestone, Sorisole, Italy) program was as follows: 5 min from room temperature to 180 °C, 10 min hold 180 °C, and 20 min vent. After cooling, the digested sample solutions were quantitatively transferred into disposable flasks and diluted to 100 mL with deionised water produced by the water Purelab DV35 purification system (ELGA, Lane End, UK). Operating conditions of the instrument were as follows: RF power 1550 W, cooling gas flow 14 L/min, nebuliser flow 1 L/min, collision gas flow 1 mL/min, dwell time 10 ms.

Standard solutions obtained from VGH labs (Manchester, UK) were used for the qualitative analysis of the samples. All solutions (standards, internal standards and samples) were prepared in 2% nitric acid. Multielement internal standard ($^{6}$Li, $^{45}$Sc-10 ng/mL; $^{71}$Ga, $^{89}$Y, $^{209}$Bi-2 ng/mL) was introduced online by an additional peristaltic pump line covering a wide mass range. The analytical process's quality was verified by analysing the standard reference material SRM 1577c (NIST, Gaithersburg, MD, USA). The minerals were recorded as mg/100 g dry meat sample.

### 2.5. Statistical Analysis

Breasts and drumsticks were used for fatty acid and mineral analyses, collected from 20 birds per feeding group (of both genders equally), making it a total of 60 birds. The chemical analyses were performed on each meat sample (120 meat samples in total) in triplicate. All the chemical analysis data involved dietary group, gender and muscle portion as the main effects and their interactions. An analysis of variance (ANOVA) using the general linear model (GLM) procedure of the IBM SPSS Statistics 20 software (SPSS, 2010) was performed for all variables considered. If the effect of the main factor (group, gender or muscle portion) was found to be significant, a least significant differences (LSD) test was used at a 5% significance level to compare the treatment means. All the results were defined as being statistically significant at $p < 0.05$.

## 3. Results

### 3.1. Fatty Acid Composition

The *p*-values showing the impact of diet, gender, muscle portion and their interactions on the fatty acids composition of broiler meat are presented in Table 2. Of all investigated factors, the muscle portion (breast and drumstick muscles) had the most significant effect on the meat fatty acid profile of Cobb 500 chickens. There were no significant interactions ($p > 0.05$) between gender and muscle portion (G × P) and between all three main factors (D × G × P). Although all significantly affected fatty acids are presented in tables, only the primary fatty acids are further discussed.

**Table 2.** The *p*-values indicating the impact of diet, gender and portion on the fatty acid composition of Cobb 500 broiler meat.

| Fatty Acid | Diet (D) | Gender (G) | Portion (P) | D × G [1] | D × P [2] | G × P [3] | D × G × P [4] |
|---|---|---|---|---|---|---|---|
| SFA [5] | | | | | | | |
| C14:0 | 0.043 | 0.633 | 0.017 | 0.616 | 0.328 | 0.601 | 0.733 |
| C15:0 | 0.338 | 0.271 | <0.001 | 0.368 | 0.561 | 0.735 | 0.404 |
| C16:0 | 0.093 | 0.001 | <0.001 | 0.015 | 0.724 | 0.743 | 0.231 |
| C18:0 | 0.447 | 0.616 | <0.001 | 0.330 | 0.705 | 0.567 | 0.531 |
| MUFA [6] | | | | | | | |
| C16:1 n-7 | <0.001 | 0.004 | <0.001 | 0.001 | 0.533 | 0.529 | 0.328 |
| C17:1 | 0.503 | 0.349 | <0.001 | 0.803 | 0.912 | 0.492 | 0.593 |
| C18:1 n-9 | 0.037 | <0.001 | <0.001 | 0.149 | 0.474 | 0.161 | 0.486 |
| C18:1 n-11 | 0.313 | 0.585 | <0.001 | 0.101 | 0.534 | 0.694 | 0.992 |
| PUFA [7] | | | | | | | |
| C18:2 n-6 | 0.004 | 0.000 | <0.001 | 0.033 | 0.664 | 0.541 | 0.892 |
| C18:3 n-6 | 0.099 | 0.859 | 0.032 | 0.217 | 0.767 | 0.629 | 0.371 |
| C18:3 n-3 | 0.231 | <0.001 | <0.001 | 0.090 | 0.617 | 0.786 | 0.656 |
| C20:2 n-6 | 0.012 | 0.611 | 0.005 | 0.294 | 0.087 | 0.927 | 0.755 |
| C20:3 n-6 | 0.021 | 0.054 | 0.001 | 0.235 | 0.417 | 0.072 | 0.517 |
| C20:4 n-6 | 0.461 | 0.441 | <0.001 | 0.368 | 0.455 | 0.861 | 0.380 |
| C22:4 n-6 | 0.868 | 0.940 | <0.001 | 0.992 | 0.907 | 0.950 | 0.414 |
| C22:6 n-3 | 0.789 | 0.028 | 0.005 | 0.085 | 0.735 | 0.729 | 0.621 |
| Total | | | | | | | |
| SFA | 0.183 | 0.004 | <0.001 | 0.106 | 0.780 | 0.924 | 0.445 |
| MUFA | <0.001 | <0.001 | <0.001 | 0.152 | 0.455 | 0.298 | 0.994 |
| PUFA | <0.001 | <0.001 | 0.004 | 0.006 | 0.656 | 0.639 | 0.490 |
| n-6 | <0.001 | <0.001 | 0.039 | 0.013 | 0.605 | 0.608 | 0.321 |
| n-3 | 0.249 | <0.001 | <0.001 | 0.014 | 0.721 | 0.692 | 0.496 |
| n-6/n-3 [8] | 0.036 | 0.005 | <0.001 | 0.285 | 0.421 | 0.277 | 0.212 |

[1] D × G—Interaction between diet and gender. [2] D × P—Interaction between diet and portion. [3] G × P—Interaction between gender and portion. [4] D × G × P—Interaction between diet, gender and portion. [5] SFA—Saturated fatty acids. [6] MUFA—Monounsaturated fatty acids. [7] PUFA—Polyunsaturated fatty acids. [8] n-6/n-3—Total n-6/n-3 ratio.

The addition of nettle leaves to the broiler diet significantly affected the meat fatty acid profile (Table 3). Compared to the control and fresh nettle group, broilers fed diet containing dry nettle had higher MUFA content, mainly palmitoleic (C16:1 n-7) and oleic (C18:1 n-9) acids, and lower PUFA and total n-6 content, mainly due to the lower linoleic acid (C18:2 n-6) content. Adding nettle to the diet had no significant impact on SFA content, with the only exception being the amount of myristic acid (C14:0), which was the highest in the dry nettle (0.88%) and lowest in the fresh nettle group (0.63%), while the control group was intermediate ($p = 0.043$). Additionally, the n-6/n-3 ratio was significantly lower in the dry nettle group, with no significant difference between the other two groups.



**Table 3.** The means (±SD [1]) of the fatty acids (% total fatty acids) of Cobb 500 broiler meat significantly affected by diet.

| Fatty Acid (%) | Control | Fresh Nettle | Dry Nettle | LSD [2] |
|---|---|---|---|---|
| C14:0 | 0.69 ± 0.31 [ab] | 0.63 ± 0.21 [a] | 0.88 ± 0.22 [b] | 0.204 |
| C16:1 n-7 | 2.42 ± 0.56 [a] | 2.22 ± 0.64 [a] | 3.19 ± 1.11 [b] | 0.419 |
| C18:1 n-9 | 23.31 ± 3.21 [a] | 23.52 ± 2.75 [a] | 24.58 ± 3.37 [b] | 1.015 |
| C18:2 n-6 | 30.33 ± 3.97 [b] | 30.93 ± 3.09 [b] | 27.57 ± 4.41 [a] | 1.985 |
| C20:2 n-6 | 0.36 ± 0.30 [ab] | 0.54 ± 0.14 [b] | 0.26 ± 0.26 [a] | 0.178 |
| C20:3 n-6 | 0.42 ± 0.15 [ab] | 0.49 ± 0.21 [b] | 0.30 ± 0.23 [a] | 0.130 |
| MUFA [3] | 28.59 ± 3.12 [a] | 28.50 ± 2.72 [a] | 31.20 ± 3.90 [b] | 1.199 |
| PUFA [4] | 38.41 ± 3.70 [b] | 39.51 ± 2.87 [b] | 35.08 ± 4.78 [a] | 1.928 |
| n-6 | 35.63 ± 3.10 [b] | 36.54 ± 2.57 [b] | 32.19 ± 4.15 [a] | 1.787 |
| n-6/n-3 [5] | 13.46 ± 2.96 [b] | 12.28 ± 2.24 [ab] | 11.83 ± 2.69 [a] | 1.258 |

[a,b] Means in rows with different superscripts differ significantly at $p \leq 0.05$. [1] SD—Standard deviation. [2] LSD—Least significant difference calculated at a 5% significance level ($p = 0.05$). [3] MUFA—Monounsaturated fatty acids. [4] PUFA—Polyunsaturated fatty acids. [5] n-6/n-3—Total n-6/n-3 ratio.

All the fatty acids that significantly differed between genders are presented in Table 4. Female broilers had a higher content of total SFA (34.11%) and MUFA (30.75%) and lower total PUFA, n-6 and n-3 fatty acids compared to male chickens, which was mainly due to a significantly higher share of palmitic (C16:0), palmitoleic (C16:1 n-7) and oleic acid (C18:1 n-9). The PUFA content of female broilers was mainly affected by the lower content of linoleic (C18:2 n-6), linolenic (C18:3 n-3), and docosahexaenoic (C22:6 n-3) acids compared to males ($p < 0.05$).

**Table 4.** The means (±SD [1]) of the fatty acids (% total fatty acids) of Cobb 500 broiler meat significantly affected by gender.

| Fatty Acid (%) | Male | Female | LSD [2] |
|---|---|---|---|
| C16:0 | 21.09 ± 2.82 [a] | 23.22 ± 3.34 [b] | 1.196 |
| C16:1 n-7 | 2.35 ± 0.62 [a] | 2.87 ± 1.05 [b] | 0.342 |
| C18:1 n-9 | 22.72 ± 2.49 [a] | 24.89 ± 3.30 [b] | 0.829 |
| C18:2 n-6 | 31.71 ± 3.22 [b] | 27.51 ± 3.72 [a] | 1.621 |
| C18:3 n-3 | 2.92 ± 0.72 [b] | 2.35 ± 0.66 [a] | 0.229 |
| C22:6 n-3 | 0.35 ± 0.22 [b] | 0.20 ± 0.21 [a] | 0.132 |
| SFA [3] | 31.62 ± 4.37 [a] | 34.11 ± 4.78 [b] | 1.622 |
| MUFA [4] | 28.12 ± 2.64 [b] | 30.75 ± 3.69 [b] | 0.979 |
| PUFA [5] | 40.20 ± 2.52 [b] | 35.14 ± 4.07 [a] | 1.574 |
| n-6 | 36.98 ± 2.35 [b] | 32.59 ± 3.65 [a] | 1.459 |
| n-3 | 3.27 ± 0.68 [b] | 2.55 ± 0.62 [a] | 0.258 |
| n-6/n-3 [6] | 11.76 ± 2.59 [a] | 13.29 ± 2.58 [b] | 1.027 |

[a,b] Means in rows with different superscripts differ significantly at $p \leq 0.05$. [1] SD—Standard deviation. [2] LSD—Least significant difference calculated at a 5% significance level ($p = 0.05$). [3] SFA—Saturated fatty acids. [4] MUFA—Monounsaturated fatty acids. [5] PUFA—Polyunsaturated fatty acids. [6] n-6/n-3—Total n-6/n-3 ratio.

The fatty acids significantly affected by the variation in the muscle portion are presented in Table 5. Breast meat had a lower amount of MUFA, mainly as a result of the lower levels of palmitoleic (C16:1 n-7) and oleic (C18:1 n-9) acids and a higher content of SFA (36.67%), mainly due to a higher share of palmitic (C16:0) and stearic (C18:0) acids. The amounts of linoleic (C18:2 n-6), linolenic (C18:3 n-3), arachidonic (C20:4 n-6) and docosahexaenoic (C22:6 n-3) acids were higher in drumstick meat, which led to a higher PUFA content compared to the breast muscles. Furthermore, in this trial, drumstick meat had a higher content of total n-6 (35.56%) and n-3 (3.40%) fatty acids and a lower n-6/n-3 ratio compared to breast muscle portions ($p < 0.05$).

**Table 5.** The means ($\pm$SD [1]) of the fatty acids (% total fatty acids) of Cobb 500 broiler meat significantly affected by muscle portion.

| Fatty Acid (%) | Breast | Drumstick | LSD [2] |
|---|---|---|---|
| C14:0 | 0.63 $\pm$ 0.21 [a] | 0.84 $\pm$ 0.28 [b] | 0.167 |
| C15:0 | 3.23 $\pm$ 0.75 [b] | 1.11 $\pm$ 0.23 [a] | 0.389 |
| C16:0 | 24.48 $\pm$ 2.62 [b] | 19.84 $\pm$ 1.80 [a] | 1.196 |
| C18:0 | 8.33 $\pm$ 0.88 [b] | 7.27 $\pm$ 0.57 [a] | 0.535 |
| C16:1 n-7 | 2.16 $\pm$ 0.77 [a] | 3.06 $\pm$ 0.78 [b] | 0.342 |
| C17:1 | 0.63 $\pm$ 0.19 [b] | 0.18 $\pm$ 0.14 [a] | 0.123 |
| C18:1 n-9 | 21.27 $\pm$ 1.55 [a] | 26.34 $\pm$ 1.90 [b] | 0.829 |
| C18:1 n-11 | 2.48 $\pm$ 0.50 [b] | 1.67 $\pm$ 0.64 [a] | 0.400 |
| C18:2 n-6 | 27.58 $\pm$ 3.26 [a] | 31.64 $\pm$ 3.76 [b] | 1.621 |
| C18:3 n-6 | 0.08 $\pm$ 0.11 [a] | 0.19 $\pm$ 0.18 [b] | 0.100 |
| C18:3 n-3 | 2.05 $\pm$ 0.44 [a] | 3.22 $\pm$ 0.45 [b] | 0.229 |
| C20:2 n-6 | 0.49 $\pm$ 0.25 [b] | 0.27 $\pm$ 0.23 [a] | 0.146 |
| C20:3 n-6 | 0.50 $\pm$ 0.20 [b] | 0.30 $\pm$ 0.18 [a] | 0.106 |
| C20:4 n-6 | 4.50 $\pm$ 0.81 [b] | 2.58 $\pm$ 0.49 [a] | 0.473 |
| C22:4 n-6 | 0.86 $\pm$ 0.24 [b] | 0.47 $\pm$ 0.22 [a] | 0.179 |
| C22:6 n-3 | 0.37 $\pm$ 0.21 [b] | 0.18 $\pm$ 0.19 [a] | 0.132 |
| SFA [3] | 36.67 $\pm$ 3.07 [b] | 29.07 $\pm$ 2.29 [a] | 1.622 |
| MUFA [4] | 26.90 $\pm$ 2.08 [a] | 31.96 $\pm$ 2.52 [b] | 0.979 |
| PUFA [5] | 36.43 $\pm$ 3.94 [a] | 38.90 $\pm$ 4.21 [b] | 1.574 |
| n-6 | 34.01 $\pm$ 3.53 [a] | 35.56 $\pm$ 3.91 [b] | 1.459 |
| n-3 | 2.42 $\pm$ 0.53 [a] | 3.40 $\pm$ 0.57 [b] | 0.258 |
| n-6/n-3 [6] | 14.47 $\pm$ 2.24 [b] | 10.58 $\pm$ 1.25 [a] | 1.027 |

[a,b] Means in rows with different superscripts differ significantly at $p \leq 0.05$. [1] SD—Standard deviation. [2] LSD—Least significant difference calculated at a 5% significance level ($p = 0.05$). [3] SFA—Saturated fatty acids. [4] MUFA—Monounsaturated fatty acids. [5] PUFA—Polyunsaturated fatty acids. [6] n-6/n-3—Total n-6/n-3 ratio.

The fatty acids significantly affected by the diet and gender interaction (D $\times$ G) are presented in Table 6. The percentage of palmitic acid (C16:0) was found to be higher in female compared to male broilers, but only in the control and dry nettle group ($p < 0.05$). The percentages of both palmitic (C16:0) and palmitoleic (C16:1 n-7) acids were found to be the highest in female chickens fed diets with the addition of dry nettle ($p < 0.05$). Furthermore, in the female chickens, the share of linoleic acid (C18:2 n-6) was the lowest in dry nettle (24.06%) and highest in the fresh nettle group (30.13%), with no significant differences in this fatty acid content between male broilers. The lowest PUFA (36%) and n-6 (33%) were recorded in the meat from female broilers fed with a control diet. Interestingly, the n-3 content was the lowest in female chickens in the control (2.49%) and fresh nettle (2.26%) group and the highest in male chickens fed with dry nettle (3.52%).

**Table 6.** The means ($\pm$SD [1]) of the fatty acids (% total fatty acids) of Cobb 500 broiler meat significantly affected by the interaction between diet and gender.

| Fatty Acid (%) | Control (C) | | Fresh Nettle (F) | | Dry Nettle (D) | | LSD [2] |
|---|---|---|---|---|---|---|---|
| | Male | Female | Male | Female | Male | Female | |
| C16:0 | 21.23 $\pm$ 2.80 [a] | 23.54 $\pm$ 3.84 [b] | 21.36 $\pm$ 3.33 [a] | 21.14 $\pm$ 1.96 [a] | 20.69 $\pm$ 2.79 [a] | 24.97 $\pm$ 3.22 [c] | 1.196 |
| C16:1 n-7 | 2.32 $\pm$ 0.62 [ab] | 2.53 $\pm$ 0.52 [b] | 2.30 $\pm$ 0.50 [ab] | 2.14 $\pm$ 0.79 [a] | 2.43 $\pm$ 0.81 [ab] | 3.95 $\pm$ 0.82 [c] | 0.342 |
| C18:2 n-6 | 32.30 $\pm$ 3.76 [d] | 28.36 $\pm$ 3.34 [b] | 31.74 $\pm$ 3.27 [bc] | 30.13 $\pm$ 2.97 [b] | 31.08 $\pm$ 3.09 [bc] | 24.06 $\pm$ 1.91 [a] | 1.621 |
| PUFA [3] | 40.84 $\pm$ 2.67 [d] | 35.99 $\pm$ 2.99 [b] | 40.41 $\pm$ 3.00 [d] | 38.61 $\pm$ 2.68 [c] | 39.34 $\pm$ 2.03 [cd] | 30.82 $\pm$ 1.59 [a] | 1.574 |
| n-6 | 37.75 $\pm$ 1.99 [d] | 33.50 $\pm$ 2.51 [b] | 37.38 $\pm$ 2.69 [d] | 35.70 $\pm$ 2.36 [c] | 35.82 $\pm$ 2.22 [c] | 28.55 $\pm$ 1.13 [a] | 1.459 |
| n-3 | 3.09 $\pm$ 0.85 [bc] | 2.49 $\pm$ 0.57 [a] | 3.20 $\pm$ 0.51 [c] | 2.26 $\pm$ 0.64 [a] | 3.52 $\pm$ 0.67 [d] | 2.90 $\pm$ 0.54 [b] | 0.258 |

[a–d] Means in rows with different superscripts differ significantly at $p \leq 0.05$. [1] SD—Standard deviation. [2] LSD—Least significant difference calculated at a 5% significance level ($p = 0.05$). [3] PUFA—Polyunsaturated fatty acids.

### 3.2. Mineral Composition

The *p*-values showing the impact of diet, gender, muscle portion and their interactions on the mineral composition of broiler meat are presented in Table 7. Although all three main factors had a significant impact on the mineral composition of meat, muscle portion affected almost all analysed minerals, except Se ($p < 0.05$). Diet and muscle portion (D × P), and gender and muscle portion (G × P) interactions had significant effect on the mineral composition. In contrast, the interactions between diet and gender (D × G) and all three main factors (D × G × P) were not found to be significant.

**Table 7.** The *p*-values indicating the impact of diet, gender and portion on the mineral composition of Cobb 500 broiler meat.

| Mineral | Diet (D) | Gender (G) | Portion (P) | D × G [1] | D × P [2] | G × P [3] | D × G × P [4] |
|---|---|---|---|---|---|---|---|
| Na | <0.001 | 0.116 | <0.001 | 0.065 | 0.016 | 0.001 | 0.090 |
| Mg | 0.068 | 0.101 | <0.001 | 0.057 | 0.272 | 0.152 | 0.101 |
| K | 0.978 | 0.021 | <0.001 | 0.122 | 0.257 | <0.001 | 0.067 |
| Ca | 0.017 | 0.007 | 0.026 | 0.516 | 0.126 | 0.002 | 0.143 |
| Mn | 0.110 | 0.081 | <0.001 | 0.094 | 0.048 | 0.068 | 0.705 |
| Fe | 0.016 | 0.338 | <0.001 | 0.486 | <0.001 | 0.557 | 0.058 |
| Zn | 0.032 | 0.099 | <0.001 | 0.860 | 0.002 | 0.004 | 0.811 |
| Se | 0.001 | 0.976 | 0.074 | 0.083 | 0.516 | 0.505 | 0.063 |

[1] D × G—Interaction between diet and gender. [2] D × P—Interaction between diet and portion. [3] G × P—Interaction between gender and portion. [4] D × G × P—Interaction between diet, gender and portion.

The minerals significantly affected by the diet are presented in Table 8. Results show that adding nettle into the broiler diet significantly increased Na, Fe, Zn and Se while decreasing the Ca content. However, these effects were not consistent in both nettle groups: the dry nettle group had the highest value for Na, while supplementation with fresh nettle significantly increased the Zn content only compared to the control. Both experimental groups had significantly lower amounts of Ca, and higher amounts of Fe and Se compared to the control ($p < 0.05$).

**Table 8.** Mineral composition (mean ± SD [1]) of Cobb 500 broiler meat significantly affected by diet.

| Mineral (mg/100 g Dry Basis) | Control | Fresh Nettle | Dry Nettle | LSD [2] |
|---|---|---|---|---|
| Na | 574.81 ± 55.38 [a] | 570.49 ± 72.16 [a] | 614.34 ± 64.82 [b] | 11.156 |
| Ca | 71.47 ± 11.44 [b] | 61.43 ± 6.48 [a] | 61.52 ± 5.53 [a] | 7.630 |
| Fe | 4.43 ± 0.69 [a] | 6.23 ± 0.19 [b] | 6.72 ± 0.44 [b] | 0.536 |
| Zn | 15.56 ± 2.04 [a] | 17.79 ± 2.11 [b] | 16.27 ± 3.29 [ab] | 1.681 |
| Se | 0.05 ± 0.00 [a] | 0.09 ± 0.01 [b] | 0.10 ± 0.01 [b] | 0.027 |

[a,b] Means in rows with different superscripts differ significantly at $p \leq 0.05$. [1] SD—Standard deviation. [2] LSD—Least significant difference calculated at a 5% significance level ($p = 0.05$).

As seen in Table 9, meat from male broilers had significantly lower potassium and higher calcium content compared to meat from female birds ($p < 0.05$). The most significant difference was established for the Ca content, with male broilers having approximately 20% more Ca in meat than females.

The meat portion had the most considerable effect on the mineral content (Table 10). Drumstick meat had significantly higher contents of Ca, Na, Fe, Mn and Zn and lower Mg and K content compared to breast meat ($p < 0.05$).

**Table 9.** Mineral composition (mean $\pm$ SD [1]) of Cobb 500 broiler meat significantly affected by gender.

| Mineral (mg/100 g Dry Basis) | Male | Female | LSD [2] |
|---|---|---|---|
| K | 3413.41 $\pm$ 298.03 [a] | 3517.04 $\pm$ 163.18 [b] | 42.747 |
| Ca | 72.56 $\pm$ 9.63 [b] | 57.05 $\pm$ 6.37 [a] | 6.230 |

[a,b] Means in rows with different superscripts differ significantly at $p \leq 0.05$. [1] SD—Standard deviation. [2] LSD—Least significant difference calculated at a 5% significance level ($p = 0.05$).

**Table 10.** Mineral composition (mean $\pm$ SD [1]) of Cobb 500 broiler meat significantly affected by portion.

| Mineral (mg/100 g Dry Basis) | Breast | Drumstick | LSD [2] |
|---|---|---|---|
| Na | 486.66 $\pm$ 40.30 [a] | 679.77 $\pm$ 41.54 [b] | 9.109 |
| Mg | 450.52 $\pm$ 11.98 [b] | 365.13 $\pm$ 29.85 [a] | 6.927 |
| K | 3771.77 $\pm$ 105.11 [b] | 3158.68 $\pm$ 152.76 [a] | 42.747 |
| Ca | 61.22 $\pm$ 17.84 [a] | 68.39 $\pm$ 9.71 [b] | 6.230 |
| Mn | 0.14 $\pm$ 0.01 [a] | 0.20 $\pm$ 0.02 [b] | 0.001 |
| Fe | 3.60 $\pm$ 0.60 [a] | 7.99 $\pm$ 1.23 [b] | 0.438 |
| Zn | 8.41 $\pm$ 0.86 [a] | 24.60 $\pm$ 3.88 [b] | 1.373 |

[a,b] Means in rows with different superscripts differ significantly at $p \leq 0.05$. [1] SD—Standard deviation. [2] LSD—Least significant difference calculated at a 5% significance level ($p = 0.05$).

The minerals significantly affected by the diet and muscle portion interaction (D $\times$ P) are presented in Table 11. Feeding broilers diet supplemented with dry nettle leaves significantly increased Na content, but only in drumstick muscles ($p < 0.05$). Nettle supplementation did not significantly affect breast meat Na content ($p > 0.05$). Interestingly, the addition of nettle leaves to the broiler diet significantly increased the Fe level in the drumstick while decreasing it in breast meat ($p < 0.05$). Similarly, Zn content significantly decreased in breast muscles of the dry nettle group, while it increased in drumstick muscles of both experimental groups ($p < 0.05$).

**Table 11.** Mineral composition (mean $\pm$ SD [1]) of Cobb 500 broiler meat significantly affected by the interaction between diet and portion.

| Mineral (mg/100 g Dry Basis) | Control | | Fresh Nettle | | Dry Nettle | | LSD [2] |
|---|---|---|---|---|---|---|---|
| | Breast | Drumstick | Breast | Drumstick | Breast | Drumstick | |
| Na | 502.38 $\pm$ 54.13 [a] | 647.24 $\pm$ 22.75 [b] | 484.63 $\pm$ 20.05 [a] | 656.35 $\pm$ 19.42 [b] | 502.97 $\pm$ 18.50 [a] | 725.70 $\pm$ 30.89 [c] | 9.109 |
| Fe | 4.00 $\pm$ 0.68 [b] | 6.86 $\pm$ 0.94 [c] | 3.38 $\pm$ 0.47 [a] | 8.39 $\pm$ 1.12 [d] | 3.42 $\pm$ 0.51 [a] | 8.22 $\pm$ 0.27 [d] | 0.438 |
| Zn | 9.13 $\pm$ 0.61 [b] | 21.79 $\pm$ 2.51 [c] | 8.94 $\pm$ 0.16 [b] | 27.14 $\pm$ 4.42 [d] | 7.48 $\pm$ 0.22 [a] | 25.87 $\pm$ 2.85 [d] | 1.373 |

[a–d] Means in rows with different superscripts differ significantly at $p \leq 0.05$. [1] SD—Standard deviation. [2] LSD—Least significant difference calculated at a 5% significance level ($p = 0.05$).

The minerals significantly affected by the gender and muscle portion interaction (G $\times$ P) are given in Table 12. Data in Table 8 shows that meat from male chickens had a higher Ca and lower K content than females, which significantly differed only for breast and drumstick muscles, respectively ($p < 0.05$). It was also found that male chickens had a significantly higher Na content than females, but only in breast muscles ($p < 0.05$). The level of Zn was the highest in male drumstick muscles ($p < 0.05$), with no significant differences between genders in breast meat.

**Table 12.** Mineral composition (mean $\pm$ SD [1]) of Cobb 500 broiler meat significantly affected by the interaction between gender and portion.

| Mineral (mg/100 g Dry Basis) | Male | | Female | | LSD [2] |
|---|---|---|---|---|---|
| | Breast | Drumstick | Breast | Drumstick | |
| Na | 502.80 $\pm$ 37.55 [b] | 675.06 $\pm$ 56.66 [c] | 470.52 $\pm$ 38.13 [a] | 684.47 $\pm$ 20.17 [c] | 9.109 |
| K | 3791.29 $\pm$ 96.38 [c] | 3035.53 $\pm$ 78.00 [a] | 3752.24 $\pm$ 115.46 [c] | 3281.84 $\pm$ 96.85 [b] | 42.747 |
| Ca | 74.23 $\pm$ 16.61 [c] | 70.88 $\pm$ 7.55 [bc] | 48.20 $\pm$ 4.39 [a] | 65.90 $\pm$ 11.38 [b] | 6.230 |
| Zn | 8.30 $\pm$ 0.92 [a] | 26.61 $\pm$ 3.99 [c] | 8.53 $\pm$ 0.82 [a] | 22.59 $\pm$ 2.64 [b] | 1.373 |

[a–c] Means in rows with different superscripts differ significantly at $p \leq 0.05$. [1] SD—Standard deviation. [2] LSD—Least significant difference calculated at a 5% significance level ($p = 0.05$).

## 4. Discussion

### 4.1. Fatty Acid Composition

The fatty acid profile of monogastric animals is majorly dependent on the diet, as the fatty acids present in the feed are incorporated more directly into animal tissues [31]. As the feed in this trial was based mainly on corn and soybean, which are generally rich in oleic (C18:1 n-9) and linoleic (C18:2 n-6) acids [32], they were the most abundant acids in all analysed samples. Together with palmitic acid (C16:0), these two fatty acids accounted for more than 75% of total fatty acids, which is in agreement with previous studies on chicken fatty acid profile [33,34].

Even though the previous research shows that nettle leaves contain polyunsaturated fatty acids, of which linolenic acid is the most abundant, followed by palmitic and linoleic acids [18,35], the content of these fatty acids, in the present study, was not increased in Cobb 500 broilers fed with nettle leaves (Table 3). The results of this trial show that the addition of dry nettle influenced higher MUFA and lower PUFA content (Table 3) and are in disagreement with the trials performed on pigs by Szewczyk et al. [36] and Hanczajowska et al. [37], where the use of nettle extract lowered meat palmitoleic acid (16:1 n-7) and total MUFA content and raised linoleic acid (C18:2 n-6) and total PUFA content. As the polyphenol content of nettles can broadly impact the lipolytic effects and thus the fat amount and quality [38], the findings of Hossain et al. [39] showing that, by drying herbs, the amounts of these phenolic compounds increase, can be used as an explanation for a more significant impact of dry nettle leaves on the meat fatty acid profile obtained in this study.

All n-6/n-3 ratios established were above the recommended requirements (<5) concerning human health [40]. Similarly, Chen et al. [33] and Costa et al. [34] report even higher n-6/n-3 ratios for breasts and thighs, above 40. A reduced n-6/n-3 ratio of meat from broilers fed with nettle leaves is also reported by Đukić Stojčić et al. [28]. However, the same authors have found that nettle supplementation increases the n-3 PUFA of breast muscles, which has not been observed in the present trial. This discrepancy between studies and a relatively large n-6/n-3 ratio can be majorly attributed to a different feed used in each experiment [41].

The variations in the meat fatty acid profile between genders (Table 4) are in agreement with the findings of Baeza et al. [42], showing that the most significant difference is that compared to males, female birds generally have a higher SFA content, which is correlated to their higher fat deposition in peripheral tissues. In this trial, higher levels of linolenic acid (C18:3 n-3), as a primary precursor for other long-chain PUFAs [43], were the main reason for the significantly lower n-6/n-3 ratio of males compared to female chickens (Table 4).

It has previously been reported that the fatty acid composition of phospholipids and neutral lipids (triacylglycerols) depends on the muscle fibre type [44–46]. Muscles, such as drumsticks, have a higher triacylglyceride content, which is rich in MUFAs, especially oleic acid [47]. These fatty acid differences between muscles are confirmed in this study (Table 5). Baeza et al. [42] report similar results, finding higher MUFA and lower SFA content in chicken thighs compared to breast. As the chicken breasts have a higher content of PUFA reached phospholipids in their membranes [44,45], it is generally assumed that breast

muscles will contain more total PUFA compared to thighs or drumsticks [46]. Contrary, in this study, the amount of PUFAs was higher in drumsticks than in breast muscles (Table 5). The explanation of these opposite findings can be found in the work of Geldenhuys et al. [46], that the phospholipid fraction has a minor influence on the overall fatty acid profile, especially in monogastric animals, where it is mainly determined by the amount and composition of neutral lipids (triacylglycerols). Furthermore, the lower n-6/n-3 ratio compared to breast muscle portions (Table 5) makes drumsticks a healthier choice for consumption [48].

Although the effect of nettle on lipid metabolism was not assessed in the present research, the results suggest that the meat fatty acid profile can be improved by adding dry nettle leaves, especially to the diet for male broilers (Table 6). However, further research is required to understand better the mechanisms that *Urtica dioica* supplementation has on lipid metabolism.

### 4.2. Mineral Composition

In the present study, potassium was the most abundant mineral in broiler meat, followed by sodium, which is in agreement with the findings of Majewska et al. [49] and Chen et al. [33].

The addition of nettle leaves to broiler diet improved the mineral profile of meat by increasing the level of Fe, Zn and Se (Table 8), which have been reported to have many beneficial effects on human health [50]. In a similar trial, feeding Ross 308 broilers with a diet containing 0.25% of rosemary leaves increased Ca and decreased Na, Mg and Fe content of meat [51]. Feeding broilers with rosemary leaves had almost the opposite effect on meat mineral composition compared to the nettle supplementation in the current experiment. Adhikari et al. [52] show that nettle leaves (*Urtica dioica*) are rich in minerals, especially calcium (169 mg/100 g oven-dry nettle leaves) and iron (277 mg/100 g oven-dry nettle leaves). In this trial, the meat Fe content increased when broilers were fed with nettle; however, the Ca content was significantly decreased ($p < 0.05$). This reduction in Ca level can be due to a higher Ca accumulation in bones, as nettle is reported to improve bone health [53,54]. However, further trials should be conducted to confirm this claim.

In the investigation of the meat mineral composition of three Chinese chicken breeds (crossbreed chicken ([817]C), Arbor Acres broiler and Hyline brown), Chen et al. [33] have found higher Ca, Na, Zn and Fe and lower K and Mg content in thigh compared to breast meat, which is in agreement with results of this study (Table 10). The same authors report lower values for Ca (from 17.82 to 31.53 mg/100 g dry weight), K (from 953.14 to 1477.78 mg/100 g dry weight) and Na (from 127.34 to 175.52 mg/100 g dry weight), probably due to the different feed used, as it will significantly impact meat mineral composition. On the other hand, Chen et al. [33] reported similar values for Fe and Zn, compared to the data presented in this trial. Due to their beneficial effect on the diet [50], the higher levels of Fe and Zn in drumstick suggest that this muscle portion has a more favourable mineral composition than breast meat. However, as higher iron levels are associated with an increase in metallic or livery meat flavour [55], further studies are needed to test the effects of nettle supplementation on meat sensory profile.

With the findings of Bao et al. [56] showing that Zn deficiency can decrease the feed intake and body growth of Cobb broilers, and the previously reported effects of nettle supplementation on increasing the growth performance and carcass weight of chickens [9,19,22,23], the results of this research propose that this weight increase is probably more pronounced in drumstick muscles, as a result of higher Zn content (Table 11).

Considering the economic and environmental impact, the poultry animal feed industry must bring an effective, natural and cheap product to market. This research shows that nettle leaves were a useful additive in broiler feed, influencing better meat quality with improved fatty acid and mineral content. As this study investigated only one nettle concentration and feeding time, the optimal dosage/time ratio remains to be investigated. In



addition, further studies are also needed to explain the mechanism of bioactive components of nettle on the meat quality characteristics.

## 5. Conclusions

The data obtained in this study suggest that feeding Cobb 500 broiler with the addition of nettle significantly influenced meat fatty acid and mineral composition, and it was strongly linked to gender and meat portion of broilers, as well as their interactions. Adding dry nettle to the diet significantly affected the meat fatty acid profile, while adding either nettles leaves significantly affected the meat mineral profile. Dry nettle supplementation had different effects in male and female chickens; namely, it decreased the PUFA and n-6 content in females while increasing the n-3 content of male chickens. Similarly, feeding broilers with the addition of nettle leaves significantly increased Fe and Zn content in drumsticks, but not in breast muscles.

**Author Contributions:** Conceptualization, N.S., M.L. and Z.Š.; methodology, N.S., D.M., M.L. and V.P.; writing—original draft preparation, N.S., M.L. and Z.Š.; writing—review and editing, N.S., D.M., M.P. and M.G.; investigation, D.M., M.P. and M.G. All authors have read and agreed to the published version of the manuscript.

**Funding:** This research was funded by the Ministry of Science, Technological Development and Innovation of the Republic of Serbia, on the basis of the Agreement on the realization and financing of scientific research work of SRO No. 451-03-47/2023-01/200022.

**Institutional Review Board Statement:** This trial was reviewed and approved (no. 05-8187/04 October 2021) by the Ethics Commission of the Institute for Animal Husbandry, Belgrade. The experiment complied with the principles of the Serbian Law 41/2009 concerning animal welfare and Rulebook 39/10 for the handling and protection of animals used for experimental purposes, as well as the EU Council Directive 98/58/EC concerning the protection of farmed animals and Directive 2010/63/EU on the protection of animals used for scientific purposes.

**Data Availability Statement:** Not applicable.

**Conflicts of Interest:** The authors declare no conflict of interest.

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
