# Peer review of "The Fatty Acid and Mineral Composition of Cobb 500 Broiler Meat Influenced by the Nettle (Urtica dioica) Dietary Supplementation, Broiler Gender and Muscle Portion"

_agriculture, doi:10.3390/agriculture13040799_

Round 1

Reviewer 1 Report

Manuscript ID: agriculture-2301020

Title:  The fatty acid and mineral composition of Cobb 500 broilers meat as affected by nettle (Urtica dioica) supplementation, gender and portion

The Manuscript is logically structured, well-illustrated and documented, contains extensive information, is written in good language and is easy to read.

The following can be noted as comments:

The discussion should add the limitations of the study and future directions, as well as discuss the results and explain the practical implications of the work. The authors should discuss the results clearly and should provide relevant information.

It is preferable to shorten the "5. Conclusions"

Lines 25-27: Rewrite this sentence

Line 53: "Urtica dioica" should be italic

Line 58: Add "were" before "significantly decreased"

Line 89: " ad libitum " should be italic

Line 87: What’s the length and humidity of the straw?

In table 1: What were the texture and size of the Mycotoxin binder (zeolite)?

Line 14: all birds were slaughtered & Line 107: twenty birds per diet group were slaughtered. which is correct? Did you slaughtered all the birds or slaughtered only 20 birds?

Lines 117 &118: what does this chemical structure mean? Check it out " (MeOH)"

Line 163: Why did you use the a least significant differences (LSD) test to compare means?

In Tables 3-6 & 8-12: what does this chemical percentage "0.5%" mean before LSD?

In Table 9: added other Mineral composition means, even if it is not significant. Why 2 only?

Lines 385: Change " increased " to "was increased "

Line 386: Add "were" before " significantly decreased "

Author Response

We would like to thank the reviewer for his/her comments. Our manuscript has been corrected. Please see the attachment.

Reviewer 2 Report

1 in paper title,what is “portion“? "musle portion" is more accurate?

2 in Materials and Methods,the fatty acid and mineral contents of fresh or dry nettle should be determined and present? 

3 in 2.3 fatty acid analysis and mineral analysis, how many samples  and replicates were detemined? the numbers also should be presented in footnote of table 2-12.

4 in table 3, 4, 6, the fatty acid name and order of the first column should be the same with Table 2 and table 5.

5 in table 8-12, the mineral name and order of the first column should be the same with Table 7.

6 there are big differences of meat mineral Na, Mg, K, Ca,Zn contents from Chen (2016), so what is the reason? should try to analyse them.

Author Response

We would like to thank the reviewer for his/her comments. Our manuscript has been checked by a native English-speaking colleague and corrected. Please see the attachment. 
